# Determinants of adult sedentary behavior and physical inactivity for the primary prevention of diabetes in historically disadvantaged communities: A representative cross-sectional population-based study from Reunion Island

Adrian Fianu[1,2]*, Sylvaine Jégo[3]◉, Christophe Révillion[3]◉, Victorine Lenclume[2], Lola Neufcourt[1], Fabrice Viale[4‡], Nicolas Bouscaren[2], Sylvain Cubizolles[4‡]

1 Centre d'Epidémiologie et de Recherche en Santé des POPulations (CERPOP), Institut National de la Santé et de la Recherche Médicale (INSERM), Université Paul Sabatier, Université de Toulouse, Toulouse, France, 2 INSERM Centre d'Investigation Clinique CIC1410, Centre Hospitalier Universitaire de La Réunion, Saint-Pierre, La Réunion, France, 3 Espace-Dev, IRD, Univ La Réunion, Saint Denis, La Réunion, France, 4 Espace-Dev, Univ La Réunion, Saint Denis, La Réunion, France

◉ These authors contributed equally to this work.
‡ FV and SC also contributed equally to this work.
* adrian.fianu@chu-reunion.fr

**Data Availability Statement:** Researchers have free access to data (anonymized table in SAS,

## Abstract

Populations undergoing extensive and rapid socio-economic transitions including historically disadvantaged communities face an increased risk of type-2 diabetes (T2D). In recent years, sedentary behavior and physical inactivity have been considered modifiable determinants when developing primary prevention programs to reduce T2D incidence. Reunion Island is a French overseas department with an increasing T2D population and a high level of socio-economic inequality. The objectives of our study were to identify the individual, social, and environmental factors associated with sedentary behavior and physical inactivity among the Reunion Island adult population, and to highlight these findings in order to propose T2D primary prevention strategies aiming at alleviating local social inequalities in health (SIH). In 2021, we conducted a population-based cross-sectional telephone survey using random sampling. Participants included adults over 15 years old living in ordinary accommodation on Reunion Island (n = 2,010). Using a sequential approach, multinomial logistic regression model (explaining 3 profiles of interest: sedentary/inactive, sedentary/active, non-sedentary/inactive), and sampling-design weighted estimates, we found that 53.9% [95% confidence interval: 51.1 to 56.7%] of participants had sedentary behavior and 20.1% [95% CI: 17.8 to 22.5%] were inactive. Abandoning physical activity due to the COVID-19 pandemic (p<0.001), final secondary school diploma or above (p = 0.005), student as professional status (p≤0.005) and living in fewer poor neighborhoods located far from city centers (p = 0.030) were four conditions independently associated with sedentary/inactive and/or sedentary/active profiles. Based on these findings, to help reduce SIH, we

SPSS, STATA and CSV formats) and explanatory documents (questionnaire, dictionary of variables). Data can be found at: https://data.progedo.fr/studies/doi/10.13144/lil-1584 The procedure for accessing the data is automatic, subject to signing a standard scientific use agreement (confidentiality, ethics, non-commercial use, citation).

**Funding:** This research was funded by the European Regional Development Fund (PO IV 2014-2020), the Regional Council of La Reunion and the French State. The funders had no role in the design of the study; in the collection, analyses, or interpretation of data; in the writing of the manuscript, or in the decision to publish the results.

**Competing interests:** The authors have declared that no competing interests exist.

used a typology of actions based on the underlying theoretical interventions including four main action categories: strengthening individuals (using person-based strategies), strengthening communities, improving living and working conditions, and promoting health-based macro-policies. Our findings suggest several directions for reducing lifestyle risk factors and enhancing T2D primary prevention programs targeting psychosocial, behavioral, and structural exposures.

## Introduction

Populations undergoing extensive and rapid socio-economic transitions including historically disadvantaged communities, are exposed to contextual changes that influence health-related behaviors [1]. Literature has shown that such changes increase social inequalities in health (SIH) and have resulted in the emergence of chronic diseases often referred to as 'first-world diseases' that are intrinsically linked to both the development of a sedentary lifestyle and excessive caloric intake through metabolic disturbances (insulin resistance) [2, 3].

Type-2 diabetes (T2D) is the predominant form of diabetes and a global public health concern [4, 5]. Modifiable determinants most frequently recognized as being responsible for T2D are those related to lifestyle (both individual and collective) [6, 7]. However, T2D can also be linked to contextual influences (i.e., non-individual health determinants), such as macroeconomic, cultural and environmental conditions prevailing in a country or a region, conditions in which people live and work, in particular, material resources for exercise and health promotion within communities [8]. Low socio-economic populations tend to have a higher prevalence of T2D [9], and historically disadvantaged communities [10] such as PIMA Indians of Arizona are particularly at risk of T2D. In these vulnerable populations, two combined mechanisms could explain the increased T2D prevalence: first, the Lifestyle Westernization [8] due to a rapid socio-economic transition that impacted the health-related behaviors, such as dietary habits (with more junk food eating) and physical and/or sporting activities (PSA); second, the interaction between environmental changes towards plenty periods and population genetic susceptibility [11, 12].

On Reunion Island, a French overseas department of 860,000 inhabitants located in the SouthWestern Indian Ocean (SWIO) region within the World Health Organization (WHO) African region, around 40% live in poverty and T2D affects over 10% of the population [9, 13, 14]. This multiethnic and multicultural community faces a long history of social inequality driven by centuries of slavery and indentured labor [13]. Despite such a painful historical context, this postcolonial society has developed a capacity of 'Living together' [15], which could help to design public health interventions based on local-networks. Nowadays, social inequalities on Reunion Island are linked to significant differences in income and to unfavorable living and working conditions [14] primarily inherited from the colonial period (18th-20th centuries) [15]. Even though the island has developed a modern healthcare system commensurate with that available in mainland France, it has not been enough to reduce the risk in T2D [13]. This observation suggests the deleterious effects of historical events (namely, servitudes) contributing to social inequalities which still expose the local population to increased health risks. According to the European Health Interview Survey (EHIS), 23% of the island's population over 15 years of age who live at home have an excessively sedentary lifestyle [16]. Sedentary behavior is defined as the amount of time spent sitting or lying down when awake, during passive transport, leisure or during professional activities [17]. It is a form of physical inactivity

with a minimal energy expenditure close to that of resting basal metabolic rate. As a demographic study from Reunion Island suggests [18], sedentary behavior is even more pronounced in territories with an ageing population. Given the abovementioned, we hypothesized that the knowledge of behavioral and health characteristics as well as the role played by the living conditions and residential environmental factors on sedentary behavior and physical inactivity could contribute to the design of T2D primary prevention programs while also taking local SIH into consideration.

To help reduce SIH, a useful typology of actions based on the underlying theoretical interventions includes four main action categories: strengthening individuals (using person-based strategies), strengthening communities, improving living and working conditions, and promoting health-based macro-policies [19]. The objectives of our study were to identify the individual, social, and environmental factors associated with sedentary behavior and physical inactivity among the Reunion Island adult population, and to highlight these findings in order to propose T2D primary prevention strategies aiming at alleviating local SIH.

## Materials and methods

### Study design and settings

A telephone population-based cross-sectional study was conducted from September 6 to December 3, 2021, using random sampling. Participants were included if they were ≥15 years old, and were residents living in ordinary accommodation on Reunion Island in 2020. Self-reported data from this regional survey (known as ERPPS or *Enquête Régionale sur les Pratiques Physiques et Sportives*) concerning the practice of PSA on Reunion Island had been collected [20]. This French public statistical survey was aimed to describe the PSA for the four administrative micro-regions (northern, eastern, southern, and western regions) on Reunion Island [20]. The descriptive results were published formerly in a study report for public health stakeholders, academics and funders [21] but not as a scientific publication. The present study is original research conducted from a secondary analysis of the ERPPS database.

The ERPPS survey was a modified replica of the nationwide study of PSA (ENPPS-2020) administered by the French national Institute for Youth and Popular Education (INJEP), with the same general objective and methodology (sampling frame, sampling methods and questionnaire). The ENPPS-2020 aimed to describe in France all PSAs, whether autonomous or supervised, to reveal the diversity of PSAs (such as emerging practices), to estimate the number of license-holder members, and to characterize the socio-economic profile of participants and non-participants (comprehensive study description available at: https://data.progedo.fr/studies/doi/10.13144/lil-1620). Identical to the ENPPS-2020, the sampling used for the ERPPS survey employed the table of individuals for 2020 compiled from a demographic listing (for tax purposes) of accommodation and individuals (Fidéli or *Fichiers démographiques sur les logements et les individus*) produced by the French National Institute for Statistics and Economic Studies (INSEE).

Individuals were directly selected through a one-level systematic random sampling stratified by the micro-region (northern, eastern, southern and western), age range (15–19, 20–29, 30–49, 50–59, 60–69 years and ≥70 years), sex (male/female) and level of income (by decile). In practice, the first step was to cross-tabulate the micro-region and age of individuals (sampling unit), leading to a sample frame of 24 strata sorted by sex and income level. The second step was to apply probability sampling. The youngest and oldest age groups (less represented in the Fidéli table than in the overall population) were over-sampled. Residents in the less densely populated (eastern region) were over-sampled compared to the other three micro-regions. The total sampling probability was unequal between the strata.

| | | SEDENTARY STATUS | |
|---|---|---|---|
| | | Sedentary | Non-sedentary |
| **PHYSICAL ACTIVITY STATUS** | Inactive | **Sedentary/Inactive** *Category presenting the highest risk for health* | **Non-sedentary/Inactive** |
| | Active | **Sedentary/Active** | **Non-sedentary/Active** *Protective category (reference)* |

**Fig 1. The four risk profiles based on sedentary and physical activity status.**

## Data collection

Individual self-reported data were gathered during the study period via a 35-minute telephone questionnaire (S1 Appendix) by native Creole-speaking interviewers recruited by the Ipsos Observer survey company. The questionnaire collected data on socio-demographic characteristics, PSA as well as other health-related behaviors, respondent's state of health, psychosocial factors, living conditions, and place of residence. Ecological data concerning the residential environment of the participants were also compiled (see Factors of exposure).

## Lifestyle behaviors outcomes

We took four risk profiles into consideration (Fig 1) including sedentary behavior and physical inactivity.

Sedentary behavior was a binary variable defined by the presence of at least one of the two following situations: i) being frequently or constantly seated in the context of their main activity during a typical day, and/or ii) using a screen for at least three hours per day outside working or study hours (including weekdays and weekends). The ERPPS survey was unable to apply the WHO's definition for recommended physical activity [17], therefore we applied proxies on the frequency of practice in three different contexts (domestic, professional and recreational) for the entire target population. Thus, physical inactivity was a binary variable which combined four reported outcome variables in the context of an individual's daily activity: i) walking only occasionally or never; ii) occasionally or never carrying or transporting loads; iii) occasionally or never practicing PSA linked to their main activity; iv) having practiced PSA no more than once a week during the previous 12 months, or not having provided information concerning such activity.

## Confounding factors

Both age range (in quartiles: 15 to 29, 30 to 44, 45 to 59 years, and 60 years and over), and sex (male/female) were potential confounding factors [22] regarding the relationship between sedentary behavior or physical inactivity and the state of health. A positive perception of the personal and parental history of PSA (Yes/No) was a potential confounding factor regarding the relationship between sedentary behavior or physical inactivity and individual socio-economic status. S2 Appendix details the data management and rationale for the latter factor.

## Factors of exposure

A contextual-dependent behavioral characteristic was defined as at least one PSA having been completely abandoned as a result of the COVID-19 pandemic (Yes/No or not concerned). Health characteristics included: the perception of their general state of health (very good, good, quite good, bad to very bad), having disability (Yes/No) and self-reported body mass index (BMI). The BMI distribution was divided into four WHO defined classes: underweight ($<18.5$ kg/m$^2$), normal (18.5–24.9 kg/m$^2$), overweight (25.0–29.9 kg/m$^2$) and obese ($\geq30.0$ kg/m$^2$) [23]. The language spoken in the home at the age of five (Only Creole/Only French/French and Creole/Other languages) was assumed to proxy literacy and community belonging. The living conditions were defined as: i) composition of household (Living alone/single living with others/Living in a couple with others), perception of financial difficulties of the household (Good or comfortably off/Just breaking even or in difficulty), the highest diploma obtained (No diploma or primary level of education/Lower high-school education or professional certificate/Secondary school diploma or above) and the professional status (Employed/Unemployed/Student/Retired/Other). The category 'Other' in the professional status included those staying at home and not on parental leave, long-term sick leave, the disabled and categories of inactive people (other than unemployed, student or retired).

We linked the residential environment to individual ERPPS data by applying the geo-localization of the participant's residential address. The first physical environment factor was the rate of the artificial coverage (buildings, parking spaces and roads) of the aggregated units for statistical information (IRIS) corresponding to an area comprising of a mean 2,500 inhabitants ($\leq 36\%$ or $>36\%$). The second physical environment factor was the mean annual temperature of IRIS residence over 30 years (1987 to 2017) classified into three groups ($\leq21.3$˚C, 21.4 to 22.4˚C, $>22.4$˚C). Additional explanations of these two indicators are available in S2 Appendix. The socio-economic level was categorized 1 to 5 on the homogeneous deprivation group of the INSEE typology [24]: 1 = urban neighborhoods facing multiple socio-economic difficulties, 2 = predominantly rural neighborhoods inhabited by poor homeowners, 3 = vulnerable neighborhoods located close to city centers, 4 = less poor neighborhoods located far from city centers, and 5 = better off neighborhoods.

## Study size

The ERPPS database secondary analysis was made *a posteriori* (i.e., defined and conducted after the inclusion and data collection). Stratified (e.g., on micro-region) multivariable regression analyses were not anticipated in the study design. Given this framework, the present study did not match the sample size pre-specification and the study size was determined by a fixed available sample of individuals [25].

## Statistical analysis

Descriptive statistics included prevalence estimate (%) and the 95% confidence interval (95% CI). Comparisons between independent samples used the Rao-Scott's chi-square test for categorical variables.

A multinomial logistic regression model was adapted to the categorical dependent variable (the risk profile of the sedentary behavior and physical inactivity in four categories). In this model, each of the three categories of interest was compared to the "Non-sedentary/Active" category of reference (see Fig 1) using an adjusted odds ratio (aOR) and the 95% CI. The aOR measured the statistical association between the risk profile category of interest and each factor of exposure, independent of the other factors. For the professional status, the reference category corresponded to the lowest prevalence of sedentary behavior in the contingency table.

The selection strategy of factors was performed using a theoretical framework for the organization of health determinants commonly mobilized in the context of social epidemiology according to a socio-ecological perspective [26, 27]. This framework was previously applied in O'Donoghue et al. (2016) [28]. Nested multivariable regression models estimated using the same statistical sample (number) were built into the application in the following sequence (see S1 Fig).

Uninterpretable data (do not know/undeclared) of low frequency (<1.7%) were the object of modalities grouping or simple imputation for three self-reported factors of exposure. Simple imputation was based on a binary logistic regression model that included complete potentially explicative variables under the missing-at-random assumption. The total non-response (people who could not be contacted, refused to participate or abandoned before the end of the telephone interview) was treated firstly by data reweighting using the homogenous response groups method, then data calibration using the CALMAR procedure (including factors associated with study participation) to increase regional representativeness of PSA [29]. All statistical analyses took the ERPPS study sampling design into account (including unequal sampling probability between the strata) and the results were weighted accordingly. The statistical analysis was carried out using the *PROC SURVEY* procedures (SAS software, Version 9.4 TS1). Statistical significance level was set to 5% and all tests were two-tailed.

## Ethical considerations

Our study was conducted in accordance with the Declaration of Helsinki and the French law No. 51-711, June 7, 1951, related to statistical obligation, coordination and privacy. The study obtained ethical, methodological, and public statistical approval by the French National Council for Statistical Information (under Visa number 2021X08IAU) and the French Ministry of the Economy, Finances and Recovery (decree dated June 18, 2021). This study respected the General European Regulation 2016/679 (April 27, 2016) concerning the protection of data (GDPR) and Law No. 78-17 (January 6, 1978) about information technology, data files and personal liberty. Answers to the questionnaire were protected by secrecy legislation. The University of La Reunion, the Regional Council of Reunion Island and the service provider responsible for collecting data were the sole recipients of identification data (name and contact details). This data was kept by the service provider for the duration of the data collection period and for an additional 12 months.

The ERPPS study was a *telephone* population-based survey. Therefore, informed consent was verbal instead of written (prior information on the study was given by postal letter and email). Study participants gave their informed verbal consent at the beginning of the interview. Obtention of informed verbal consent from the parent or guardian was a prerequisite for the interviewer before obtaining informed verbal consent from the minor (15-17-year-old participant). This was approved by the French National Council for Statistical Information. All participants had the right to access, rectify, delete or limit the use of their data for up to 12 months after the end of the data collection period.

## Results

In total, 2,010 adults from Reunion Island participated in the study (Fig 2) which represented a participation rate of 49%. Participants were mostly women, aged 30 to 59 years and Reunion Island natives (data not shown).

A similar distribution of prevalence estimates was found comparing the ERPPS and Reunion Island regional statistics (Table 1). Given a prevalence confidence interval lower bound higher than 50%, over half the participants had sedentary behavior (53.9%, [95% CI:

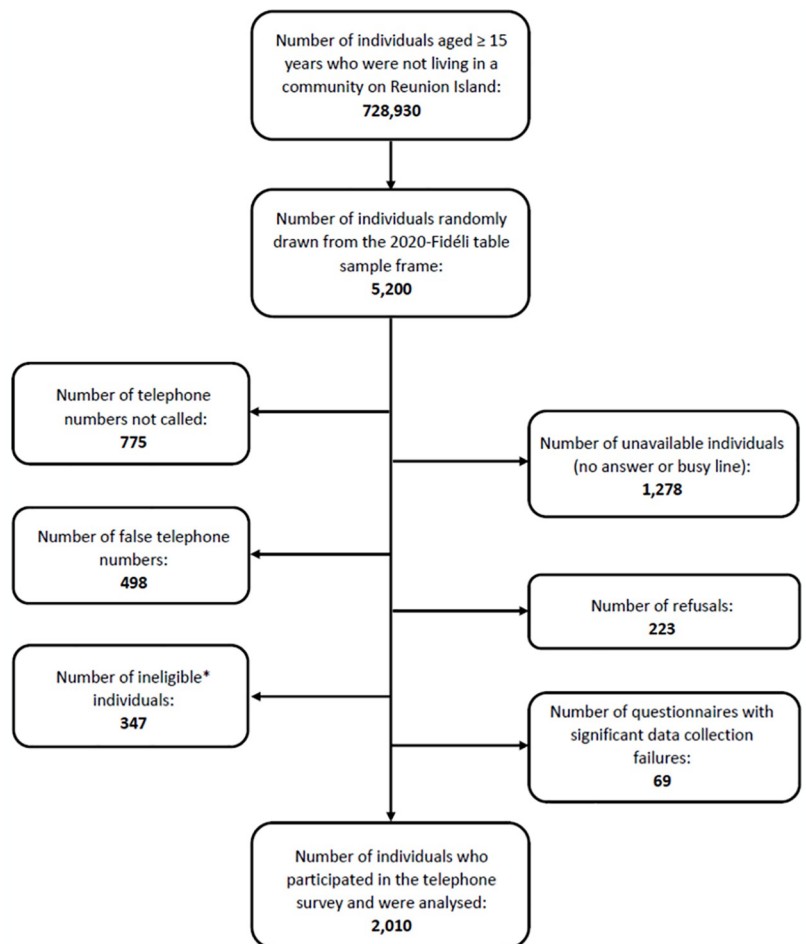

**Fig 2. ERPPS study flowchart.**

51.1 to 56.7%]). One out of five was classified as inactive (20.1%, [95% CI: 17.8 to 22.5%]). Regarding risks, 13.0% of participants presented the highest risk for health (sedentary/inactive) and 39.0% were from the protective category (non-sedentary/active).

According to administrative micro-regions, inhabitants of the northern region had the highest prevalence estimate for both sedentary/inactive (15.0%) and sedentary/active (46.1%) and the minimum prevalence estimate for non-sedentary/active (32.4%) when compared to those of the other micro-regions (Fig 3).

Participants were mostly from less poor neighborhoods located far from city centers (29.5%) (Table 1). Almost half the total was aged 15 to 44 years old (48.8%). Most participants possessed an educational degree (39.0%). Concerning attitudes and practices, most participants (79.2%) did not have a positive perception of their personal and parental history of PSA. Moreover, almost a quarter of them (23.4%) reported having completely abandoned at least one PSA as a result of the COVID-19 pandemic.

In the complete full-adjusted regression model (see M3 in Table 2), two factors that were independently associated with a sedentary/inactive profile were PSA being completely

**Table 1. Participant characteristics and regional representativeness.**

| Participant characteristics | ERPPS findings | Descriptive statistics from the Census data (INSEE) or a specific survey (EHIS) on Reunion Island |
|---|---|---|
| | % [95% CI] | % |
| **Sedentary status** | | |
| Sedentary | 53.9 [51.1 to 56.7] | - |
| Non-sedentary | 46.1 [43.3 to 48.9] | - |
| **Physical activity status** | | |
| Active | 79.9 [77.5 to 82.2] | - |
| Inactive | 20.1 [17.8 to 22.5] | - |
| **Risk profiles** | | |
| Sedentary/Inactive | 13.0 [11.2 to 15.0] | - |
| Sedentary/Active | 40.9 [38.2 to 43.7] | - |
| Non-sedentary/Inactive | 7.1 [5.7 to 8.8] | - |
| Non-sedentary/Active | 39.0 [36.3 to 41.7] | - |
| **Having practiced at least one PSA during the 12 last months** | | |
| Yes | 80.8 [78.2 to 83.2] | - |
| No | 19.2 [16.8 to 21.8] | - |
| **Age range** | | |
| 15 to 29 years | 24.0 [21.7 to 26.4] | 24.1[a] |
| 30 to 44 years | 24.8 [22.6 to 27.2] | 23.8[a] |
| 45 to 59 years | 27.2 [24.7 to 29.6] | 27.4[a] |
| 60 years and over | 24.0 [21.4 to 26.8] | 24.7[a] |
| **Sex** | | |
| Male | 46.0 [43.2 to 48.8] | 46.5[a] |
| Female | 54.0 [51.2 to 56.8] | 53.5[a] |
| **Positive perception of the personal and parental history of PSA** | | |
| Yes | 20.8 [18.7 to 23.1] | - |
| No | 79.2 [76.9 to 81.3] | - |
| **At least one practice of PSA being abandoned because of the COVID-19 pandemic** | | |
| No/not concerned | 76.6 [74.2 to 78.9] | - |

(*Continued*)

**Table 1.** (Continued)

| Participant characteristics | ERPPS findings | Descriptive statistics from the Census data (INSEE) or a specific survey (EHIS) on Reunion Island |
|---|---|---|
| | % [95% CI] | % |
| Yes | 23.4 [21.1 to 25.8] | - |
| **Perception of general state of health** | | |
| Very good to good | 59.9 [57.1 to 62.8] | 64.7[b] |
| Quite good/don't know (n = 3) | 32.5 [29.7 to 35.2] | 25.6[b] |
| Bad to very bad | 7.6 [6.1 to 9.4] | 9.6[b] |
| **Disability** | | |
| Yes | 7.2 [5.8 to 8.9] | 9.6[b] |
| No | 92.8 [91.1 to 94.2] | 90.4[b] |
| **Body mass index (missing data = 45)** | | |
| Underweight ($<18.5$ kg/m$^2$) | 6.3 [5.0 to 8.0] | 6.1[b] |
| Normal weight (18.5 to 24.9 kg/m$^2$) | 51.7 [48.9 to 54.5] | 49.3[b] |
| Overweight (25.0 to 29.9 kg/m$^2$) | 28.9 [26.4 to 31.4] | 28.4[b] |
| Obese ($\geq30.0$ kg/m$^2$) | 13.1 [11.3 to 15.2] | 16.2[b] |
| **Language spoken in the home at the age of five** | | |
| Only Creole | 38.9 [36.1 to 41.7] | - |
| Only French | 15.4 [13.6 to 17.3] | - |
| French and Creole | 39.1 [36.3 to 41.8] | - |
| Other | 6.7 [5.5 to 8.1] | - |
| **Living situation** | | |
| Living in a couple with other persons | 55.5 [52.7 to 58.3] | - |
| Living alone | 14.6 [12.6 to 16.9] | 11.5[c] |
| Single living with other persons | 29.9 [27.4 to 32.3] | - |
| **Perception of financial difficulties of the household** | | |
| Just breaking even or in difficulty | 41.1 [38.3 to 43.9] | - |
| Good or comfortably off | 58.9 [56.1 to 61.7] | - |
| **Education degree (highest)** | | |
| No diploma or primary level of education | 24.9 [22.2 to 27.8] | 38.2[d] |
| Lower high-school education or professional certificate | 36.1 [33.4 to 38.8] | 25.3[d] |

(*Continued*)

**Table 1.** (Continued)

| Participant characteristics | ERPPS findings | Descriptive statistics from the Census data (INSEE) or a specific survey (EHIS) on Reunion Island |
|---|---|---|
| | % [95% CI] | % |
| Final secondary school diploma or above | 39.0 [36.4 to 41.6] | 36.5[d] |
| **Professional status** | | |
| Employed | 40.0 [37.3 to 42.6] | 46.4[e] |
| Unemployed | 19.0 [16.7 to 21.4] | 22.8[e] |
| Student | 12.2 [10.5 to 14.1] | 11.2[e] |
| Retired | 14.0 [12.2 to 16.1] | 3.7[e] |
| Other[f] | 14.9 [12.6 to 17.6] | 15.9[e] |
| **Rate of artificial cover of the ground of the IRIS of residence** | | |
| ≤36% | 49.7 [46.9 to 52.5] | 48.7 |
| >36% | 50.3 [47.5 to 53.1] | 51.3 |
| **Mean annual temperature of the IRIS of residence over 30 years** | | |
| ≤21.3°C | 32.7 [29.9 to 35.5] | 31.1 |
| 21.4°C to 22.4°C | 33.0 [30.4 to 35.6] | 32.9 |
| >22.4°C | 34.3 [31.7 to 36.9] | 36.0 |
| **Deprivation level of the large neighborhood of residence (1: most deprived to 5: least deprived)** | | |
| 1 | 11.7 [10.0 to 13.6] | 11.8[g] |
| 2 | 21.7 [19.3 to 24.4] | 22.4[g] |
| 3 | 19.6 [17.5 to 21.9] | 19.1[g] |
| 4 | 29.5 [27.0 to 32.0] | 29.6[g] |
| 5 | 17.5 [15.6 to 19.5] | 17.1[g] |
| **Administrative micro-region of residence** | | |
| Southern region | 36.0 [33.1 to 38.9] | 36.1 |
| Eastern region | 14.0 [12.5 to 15.6] | 14.9 |
| Northern region | 26.0 [23.7 to 28.3] | 24.0 |

(*Continued*)

**Table 1.** (Continued)

| Participant characteristics | ERPPS findings | Descriptive statistics from the Census data (INSEE) or a specific survey (EHIS) on Reunion Island |
|---|---|---|
| | % [95% CI] | % |
| Western region | 24.0 [21.8 to 26.3] | 25.1 |

% [95% CI]: percentage [95% confidence interval]. PSA: physical and sporting activities. IRIS: aggregated units for the statistical information. Missing data on body mass index due to undeclared height or weight.

[a] 2021 INSEE Census data.

[b] 2019 EHIS survey data ([16]).

[c] 2019 INSEE Census data.

[d] 2019 INSEE Census data for ≥15 years old.

[e] 2019 INSEE Census data for 15 to 64 years old.

[f] Persons staying at home and not on parental leave, long-term sick leave, disabled persons and categories of inactive persons (other than unemployed, pupils, tertiary students and retired persons).

[g] Regional data ([24]).

abandoned as a result of the COVID-19 pandemic and/or the professional status of participants. Those having completely abandoned at least one practice of PSA because of the COVID-19 pandemic, and students were more likely to have sedentary/inactive profile. These associations were already significant in the first nested multivariable regression models (see M1 and M2 in S1 Table).

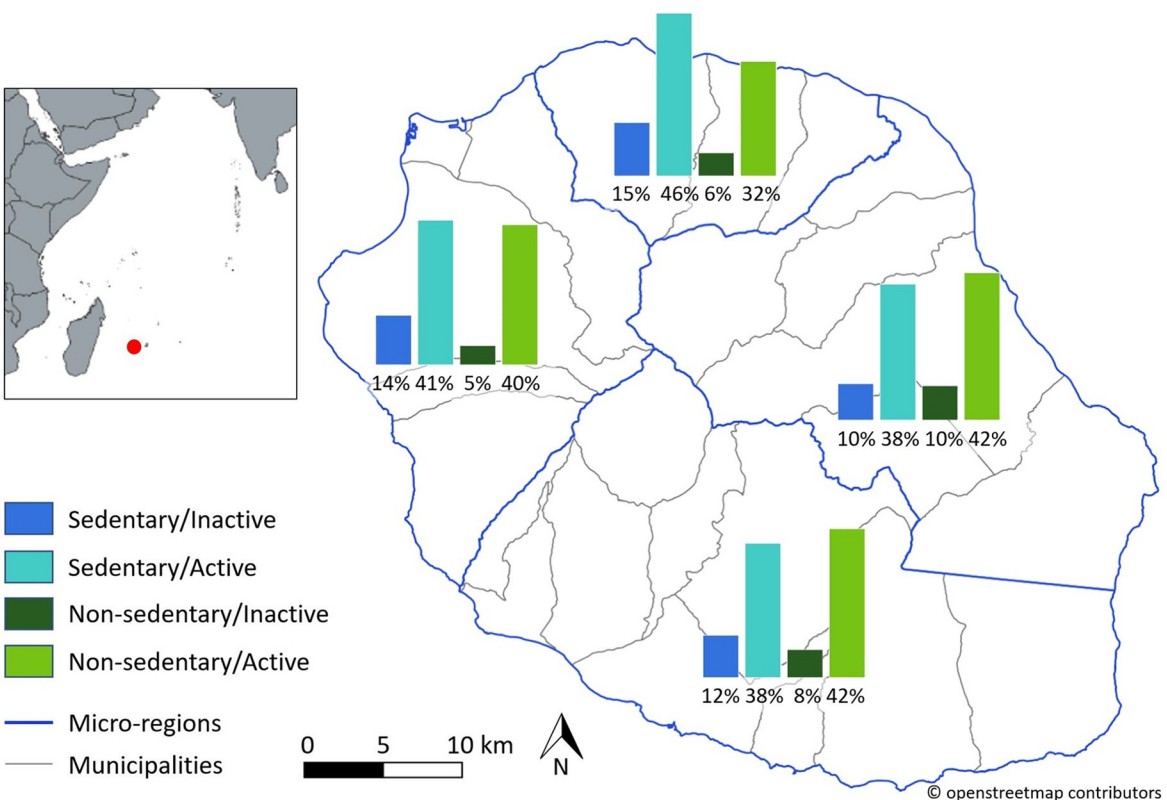

**Fig 3. Prevalence of the four risk profiles according to Reunion Island administrative micro-region.** Source: OpenStreetMap ® and ERPPS study. Map software: QGIS v3.10 (Coruna).

**Table 2. Individual and socio-environmental factors independently associated with the three risk profiles of people aged 15 years and over on Reunion Island in 2021 (M3 nested multivariable regression models using non-sedentary/active profile as reference).**

| FACTORS | Sedentary/Inactive profile | p | Sedentary/Active profile | p | Non-sedentary/Inactive profile | p |
|---|---|---|---|---|---|---|
| | aOR [95% CI] | | aOR [95% CI] | | aOR [95% CI] | |
| **At least one practice of PSA being abandoned because of the COVID-19 pandemic** | | | | | | |
| No/not concerned | 1 | - | 1 | - | 1 | - |
| Yes | 2.17 [1.43 to 3.30] | < .001 | 1.75 [1.27 to 2.41] | <0.001 | 1.67 [0.92 to 3.03] | 0.094 |
| **Education degree (highest)** | | | | | | |
| No diploma or primary level of education | 1 | - | 1 | - | 1 | - |
| Lower high-school education or professional certificate | 0.80 [0.44 to 1.45] | 0.467 | 1.10 [0.73 to 1.67] | 0.655 | 0.90 [0.47 to 1.74] | 0.752 |
| Final secondary school diploma or above | 1.47 [0.88 to 2.48] | 0.145 | 1.79 [1.19 to 2.67] | 0.005 | 0.59 [0.25 to 1.37] | 0.219 |
| **Professional status** | | | | | | |
| Other[a] | 1 | - | 1 | - | 1 | - |
| Employed | 1.32 [0.68 to 2.57] | 0.409 | 1.26 [0.76 to 2.07] | 0.374 | 0.70 [0.31 to 1.58] | 0.389 |
| Unemployed | 2.12 [0.97 to 4.65] | 0.060 | 1.30 [0.74 to 2.30] | 0.361 | 0.79 [0.35 to 1.77] | 0.558 |
| Student | 3.95 [1.51 to 10.30] | 0.005 | 4.27 [2.07 to 8.79] | <0.001 | 0.94 [0.17 to 5.30] | 0.942 |
| Retired | 0.96 [0.36 to 2.53] | 0.937 | 1.07 [0.43 to 2.69] | 0.878 | 0.51 [0.18 to 1.41] | 0.194 |
| **Rate of artificial cover of the ground of the IRIS of residence** | | | | | | |
| ≤36% | 1 | - | 1 | - | 1 | - |
| >36% | 1.21 [0.80 to 1.83] | 0.372 | 1.22 [0.87 to 1.71] | 0.244 | 1.28 [0.71 to 2.29] | 0.414 |
| **Deprivation level of the large neighborhood of residence (1: most deprived to 5: least deprived)[b]** | | | | | | |
| 1 | 1 | - | 1 | - | 1 | - |
| 2 | 1.21 [0.42 to 3.49] | 0.723 | 0.56 [0.30 to 1.07] | 0.078 | 1.20 [0.41 to 3.50] | 0.740 |
| 3 | 1.97 [0.73 to 5.31] | 0.178 | 0.61 [0.36 to 1.04] | 0.068 | 1.70 [0.65 to 4.41] | 0.278 |
| 4 | 1.59 [0.61 to 4.15] | 0.342 | 0.55 [0.32 to 0.94] | 0.030 | 0.87 [0.32 to 2.32] | 0.774 |
| 5 | 1.79 [0.70 to 4.58] | 0.228 | 0.75 [0.43 to 1.30] | 0.307 | 1.38 [0.46 to 4.11] | 0.565 |
| **Administrative micro-region of residence** | | | | | | |
| Southern region | 1 | - | 1 | - | 1 | - |
| Eastern region | 0.92 [0.52 to 1.61] | 0.767 | 0.91 [0.61 to 1.35] | 0.641 | 1.32 [0.67 to 2.61] | 0.430 |
| Northern region | 1.32 [0.76 to 2.28] | 0.323 | 1.34 [0.89 to 2.01] | 0.158 | 0.93 [0.40 to 2.15] | 0.866 |
| Western region | 1.25 [0.73 to 2.13] | 0.423 | 1.03 [0.71 to 1.49] | 0.896 | 0.85 [0.39 to 1.85] | 0.681 |

aOR [95% CI]: adjusted odds ratio [95% confidence interval]. PSA: physical and sporting activities. IRIS: aggregated units for the statistical information. All regression models were adjusted based on age range (15 to 29/30 to 44/45 to 59/60 years+), sex (female/male) and positive perception of the personal and parental history of PSA (yes/no).

[a] Persons staying at home and not on parental leave, long-term sick leave, disabled persons and categories of inactive persons (other than unemployed, pupils, tertiary students and retired persons).

[b] Typology from [24].

In the complete full-adjusted regression model (see M3 in Table 2), factors independently associated with a sedentary/active profile were: i) PSA being completely abandoned because of the COVID-19 pandemic, ii) educational qualification, iii) professional status, and iv) large neighborhood deprivation level. People having completely abandoned at least one PSA as a result of the COVID-19 pandemic, participants with an educational degree, and being students were more likely to have sedentary/active profile. Similarly, these associations were already significant in the first nested multivariable regression models (see M1 and M2 in S1 Table). Those in "less poor neighborhoods located far from city centers" (Level 4) were less likely to

have a sedentary/active profile compared to participants in "urban neighborhoods facing multiple socio-economic difficulties" (Level 1) (Table 2).

As shown in Table 2 and S1 Table, both for the complete full-adjusted regression model (see M3) and the first nested multivariable regression models (see M1 and M2), no factor was independently associated with a non-sedentary/inactive profile. Univariate associations (crude ORs) between each factor (exposures and confounders) and the 4-category outcome variable are available in S2 Table.

## Discussion

Our study confirms that there was a high prevalence of adult sedentary behavior in Reunion Island in 2021. Abandoning the practice of PSA due to the COVID-19 pandemic and the living conditions determined by education, professional status and neighborhood deprivation level were significant factors independently associated to sedentary behavior including sedentary/inactive and sedentary/active profiles.

From 2020 to 2021, the COVID-19 pandemic including the national lockdowns and other universal preventive measures based on individual and collective freedom restrictions affected the living conditions of individuals. Many studies reported the consequences on physical activity levels and sedentary behavior in general populations from this unprecedented event [30, 31]. Chêne (2022) described the situation in mainland France one year after the first lockdown [32]. In that study, ~50% of adults reported insufficient physical activity levels below the guidelines after lockdowns were lifted and this prevalence was similar to that observed during lockdown. Additionally, 21.3% of adults reported staying seated more than seven hours/day (instead of one out of three hours during lockdown) [32]. Hall et al. (2021) noted that the COVID-19 pandemic may have accelerated ongoing physical inactivity and sedentary behaviors [33]. For prevention, this collective awareness could be used by decision-makers and the public for behavioral changes towards a healthier lifestyle, taking into consideration that COVID-19 pandemic gave several examples of lockdown-related behaviors driven by the socio-economic characteristics of the population [34].

Education level and professional status are two primary indicators of an individual's socio-economic background. In 2016, a systematic review conducted by O'Donoghue et al. recognized that the socio-economic position of an individual was the most consistent correlate of adult sedentary behavior among psychological, behavioral, physical, biological and genetic factors [28]. The authors, however, noted some discrepancies depending on the measurement of sedentary behavior (subjective/objective measures) and the domain (professional/leisure times) [28]. In our study, we found that participants with a higher educational degree were more likely to exhibit sedentary and physically active periods compared to those having only a primary level education or no educational background. Rey et al. (2023) reported that a higher education degree was more often associated with a higher health literacy in French overseas departments (including Reunion Island), which may imply a better integration of communication on physical activity content [35]. Health literacy can be defined as the "skills and capacities intended to enable people to exert greater control over their health and the factors that shape health" [36]. In this way, a higher health literacy level could help a better health communication understanding to promote physical activity and decrease the harms of sedentary behavior.

The external validity assessment of the neighborhood socio-economic effect on adult sedentary behavior was challenging because of the contradictions faced [28]. In our study, inhabitants from less poor neighborhoods located far from city centers were less likely to exhibit sedentary and physically active periods compared to those from the most deprived and urban neighborhoods. People living far from city centers most likely did not have rapid access to

sporting facilities and public transport to help engage in PSA. In accordance with Paudel et al. (2023) [37], this interpretation supports the statement that physical activity inequalities could be driven by differential access to material resources at an area level. More precisely, regional planning policies can lead to inequalities in the geographical distribution of resources (such as football ground), and consequently in access to PSA for the population. For inhabitants from the most deprived living areas [24], we presumed an endogenous psychosocial mechanism (see explanation further) as shown in Kelly-Irving et al. (2021) and inhibition of some protective health-related attitudes as shown in Rey et al. (2023) which leads to a sedentary lifestyle [35, 38]. In line with the theory of planned behavior, this mechanism could imply the significant influence of subjective norm within the neighborhood on personal attitude towards the behavior [39].

Compared to other administrative micro-regions of Reunion Island, we found that northern inhabitants had both the highest risk level for sedentary behavior and the lowest likelihood for protective behaviors. The north region includes the French overseas department largest city: Saint-Denis (>150,000 inhabitants), the Reunion Island's administrative and economic chief town. This observation could be interpreted as the consequence of a rapid socio-economic transition, the end of an exclusive agricultural society, exposing people now living in urban areas to a more sedentary lifestyle [16, 18].

Socio-economic, demographic and epidemiologic transitions are often intertwined [40]. In French overseas populations, these dynamics have led to a modification of health-related behaviors and health states such as sedentary behavior, physical inactivity and T2D [41]. To describe the phenomena of sedentary behavior and physical inactivity, a framework that explains the biological mechanisms involved in social exposures embodiment was used [38]. This framework distinguishes the difference between exogenous and endogenous factors. The former comprises behavioral exposures (e.g., smoking) and structural exposures (e.g., country economic system), whereas the latter includes psychosocial exposures (e.g., experience of adversity during childhood) which are linked to personal perceptions, emotions, and stress-response system [38]. In our study, we found one exogenous behavioral exposure related to the context in which it happened (abandoning physical activity due to the COVID-19 pandemic), three exogenous structural exposures dealing with living or working conditions (education, professional status and limited access to material resources in the neighborhood), and one endogenous psychosocial exposure (inhibition of some protective health-related attitudes in the most deprived neighborhoods). This useful typology suggests several directions for primary prevention targeting structural, behavioral and psychosocial exposures. The first could be to target specific actions to populations having a high level of education and those carried out in schools and higher education. These public health actions could be implemented through processes for improving the material working and living conditions as suggested by Whitehead (2007) [19] of office workers and students such as considering the individual preferences for the organization of repeated break periods to diminish time spent sitting and to motivate people to increase movement and physical activity. The second direction could be to adapt regional policies to counteract the negative consequences of PSA reduction from the COVID-19 pandemic period. This type of intervention could be implemented through processes to promote healthy macro-policies [19] particularly for land use planning as recommended by Barry et al. (2017) such as building facilities dedicated to PSA with additional green spaces in every municipality [42]. To improve the use of these new facilities, our group suggested promoting PSA within the family unit from a very early age as well as applying a logic of education-by-peers and developing collective protective behavior to be envisioned and implemented through community strengthening processes such as local-networks [43, 44]. Another direction could be to improve the prevailing macroeconomic and environmental

conditions in neighborhoods. This could be implemented through the promotion of healthy macro-policies [19]. The main aim would be to reduce the gaps in material and psychosocial needs between neighborhoods by employing a strategy of proportionate universalism [45]. This strategy consists of allocating resources for improving relevant social determinants in proportion to the level of social disadvantage to reduce SIH. For instance, one could rank through population-based study findings the Reunion Island neighborhoods' need for community empowerment and implement differential pleasant leisure activities (e.g., walking groups, hiking, communal meals) to enhance and balance inhabitants' well-being between neighborhoods.

The understanding of individual and socio-environmental determinants of adult sedentary behavior and physical inactivity in a given place provides opportunities to propose public health interventions for the territory under study. In our case, the research perspective is to contribute to the global reflection on the designs of T2D primary prevention programs while concurrently tackling SIH. In this way, we believe that using a local investigation platform and a monitored methodological approach at the edge of social epidemiology and health geography contribute to producing relevant data for evidence-based public health practices [46]. To some extent, our findings on health determinants are not entirely transferable to other contexts and/ or populations. For instance, the neighborhood deprivation index designed on a local-based geographic entity (i.e., the large neighborhood of residence) using evolving socio-environmental indicators (e.g., the monetary poverty rate among children) is different from other indexes (such as the European Deprivation Index [47]) and specific to Reunion Island [24]. Nevertheless, our methods for sampling, data collection, and statistical analysis (the socio-ecological framework and sequential approach) could be employed in future studies. We suggest two improvements to expand the knowledge of health determinants highlighted by quantitative methods. The first is to identify the pathway between social factors of exposure, lifestyle mediators and T2D outcomes using a cohort study design [48]. The second is to add diet and nutrition in the lifestyle mediator's analysis.

Regarding the limitations of this study, adults living in institutionalized accommodations (such as retirement / assisted living facilities or long-term care facilities) were not represented in this study. This may have imposed a lack of general population coverage and an under-estimation of the sedentary behavior prevalence for the Reunion Island adult population. Secondly, we used self-reported data, instead of the accelerometer gold standard method, to measure sedentary behavior and physical inactivity. Specifically, as regards the effect of educational qualifications, we assessed the internal consistency of self-reported data. Thus, the relationship between a high level of education and the absence of sedentary behavior was unchanged when the latter indicator was replaced by the frequency of PSA$\geq$3 times a week [21]. Another limitation was that the multivariate analyses did not include all the environmental factors. Data concerning the security of the residential neighborhood (e.g., safe park), the housing characteristics (e.g., number of PCs at home) and the professional/working environment (e.g., safe bike storage at work) were lacking [28]. Future population-based studies should consider these factors in random sampling stratification and data collection to investigate their specific effects on health and decrease residual confounding due to unmeasured covariates. In the same methodological consideration, the study did not match the sample size pre-specification necessary for identifying with adequate statistical power some known risk factors. In this paper, our analytical approach was exploratory. Based on those first results, new specific hypotheses could be elaborated and assessed using an explanatory approach in future analyses. Lastly, due to the design of the study, we had difficulty checking that the factors of exposure preceded the occurrence of the behavior endpoints. However, with a cross-sectional design, the temporal condition necessary for a causal approach in epidemiology can

only be assessed for retrospective exposure factors [22], such as the education level of the oldest participants or the residential environment characteristics.

Regarding the strengths of this study, weighted estimates allowed us to describe our results at a regional level by using a population-based study with individuals drawn by random sampling methods. Moreover, data reweighting and calibration (including sex, age and place of birth) improved the study's reliability. The combination of two individual statuses at the time of the survey, namely sedentary behavior and physical activity status, including sedentary and physically inactive profile and the opposite (i.e., non-sedentary and physically active profile), was a way to rank health risks related to T2D and other non-communicable diseases such as cardiovascular diseases. Indeed, physical activity decreases the health risks from a sedentary lifestyle, however, it does not eliminate them [17]. Another strength to this study was that we used an informative questionnaire to investigate living conditions and to describe the context of sedentary behavior and physical inactivity in the island's adult population using regional environmental databases compiled from large data-gathering systems [26].

## Conclusions

As with many other historically disadvantaged communities, Reunion Island has faced significant changes in living conditions, lifestyles and the rise of T2D. From a public health perspective, our study can be used to enhance primary prevention programs aimed at reducing sedentary behavior and physical inactivity as well as equitably improve the health of the population.

## Supporting information

**S1 Appendix. ERPPS study questionnaire.**
(PDF)

**S2 Appendix. Data management and rationale.** This file gives information on composite variables: the Positive perception of the personal and parental history of PSA, and the Physical environment indicators characterizing the residential environment.
(DOCX)

**S1 Fig. Nested multivariable regression models (M0, M1, M2, and M3) sequential approach.**
(DOCX)

**S1 Table. Individual and social factors independently associated with each of the three risk profiles of people aged 15 years and over on Reunion Island in 2021 (M1 and M2 nested multivariable regression models using non-sedentary/active profile as reference).**
(DOCX)

**S2 Table. Individual and socio-environmental factors associated with the three risk profiles of people aged 15 and over on Reunion Island in 2021 (non-adjusted regression models using non-sedentary/active profile as reference).**
(DOCX)

## Acknowledgments

The authors would like to thank the INJEP for authorizing the use of the national questionnaire and supported the process of labelling the ERPPS. We acknowledge Anna Asperti and Cobie Rosewarne for English editing and internal review processes as well as Sarina Yaghobian

and Marty Brucato from AcaciaTools for their proofreading services. We also express our gratitude to the participants of the survey.

## Author Contributions

**Conceptualization:** Adrian Fianu, Sylvaine Jégo, Fabrice Viale, Sylvain Cubizolles.

**Data curation:** Sylvaine Jégo, Christophe Révillion.

**Formal analysis:** Sylvaine Jégo, Christophe Révillion.

**Funding acquisition:** Fabrice Viale, Sylvain Cubizolles.

**Investigation:** Fabrice Viale, Sylvain Cubizolles.

**Methodology:** Adrian Fianu, Lola Neufcourt, Nicolas Bouscaren.

**Project administration:** Fabrice Viale, Sylvain Cubizolles.

**Resources:** Fabrice Viale, Sylvain Cubizolles.

**Software:** Sylvaine Jégo, Christophe Révillion.

**Supervision:** Adrian Fianu, Sylvain Cubizolles.

**Validation:** Victorine Lenclume, Lola Neufcourt, Nicolas Bouscaren.

**Visualization:** Adrian Fianu.

**Writing – original draft:** Adrian Fianu.

**Writing – review & editing:** Adrian Fianu, Sylvaine Jégo, Christophe Révillion, Victorine Lenclume, Lola Neufcourt, Fabrice Viale, Sylvain Cubizolles.

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
