## [Decision Letter · Decision Letter 0]

31 May 2024

PONE-D-23-43256Determinants of adult sedentary behavior for the primary prevention of diabetes in historically disadvantaged communities: a representative cross-sectional study from Reunion IslandPLOS ONE

Dear Dr. Fianu,

Thank you for submitting your manuscript to PLOS ONE. After careful consideration, we feel that it has merit but does not fully meet PLOS ONE’s publication criteria as it currently stands. Therefore, we invite you to submit a revised version of the manuscript that addresses the points raised during the review process.

We look forward to receiving your revised manuscript.

Kind regards,

Attila Csaba Nagy

Academic Editor

PLOS ONE

 [This research was funded by the European Regional Development Fund (PO IV 2014-2020), the Regional Council of La Reunion and the French State. The funders had no role in the design of the study; in the collection, analyses, or interpretation of data; in the writing of the manuscript, or in the decision to publish the results.].  

3. We note that Figure 3 in your submission contain [map/satellite] images which may be copyrighted. All PLOS content is published under the Creative Commons Attribution License (CC BY 4.0), which means that the manuscript, images, and Supporting Information files will be freely available online, and any third party is permitted to access, download, copy, distribute, and use these materials in any way, even commercially, with proper attribution. For these reasons, we cannot publish previously copyrighted maps or satellite images created using proprietary data, such as Google software (Google Maps, Street View, and Earth). For more information, see our copyright guidelines: http://journals.plos.org/plosone/s/licenses-and-copyright.

1. You may seek permission from the original copyright holder of Figure 3 to publish the content specifically under the CC BY 4.0 license.  

Reviewers' comments:

Reviewer's Responses to Questions

**Comments to the Author**

1. Is the manuscript technically sound, and do the data support the conclusions?

Reviewer #1: Yes

Reviewer #2: Partly

2. Has the statistical analysis been performed appropriately and rigorously? 

Reviewer #1: Yes

Reviewer #2: Yes

3. Have the authors made all data underlying the findings in their manuscript fully available?

Reviewer #1: No

Reviewer #2: Yes

4. Is the manuscript presented in an intelligible fashion and written in standard English?

Reviewer #1: Yes

Reviewer #2: Yes

5. Review Comments to the Author

Reviewer #1: Manuscript Number: PONE-D-23-43256.

Title: Determinants of adult sedentary behavior for the primary pre of diabetes in historically disadvantaged communities : a representative cross-sectional study from Reunion island.

Article Type: Research article

General comment

Sedentary lifestyle is a key determinant of obesity and of its subsequent complications such as type 2 diabetes mellitus (T2D), blood hypertension, arterial stiffness, and beyond, cardiovascular ischemic diseases. Understanding the pathways between sedentary behavior risk factors, especially those that are modifiable, and T2D is therefore of paramount importance to prevent T2D and its related burden of morbimortality. In this framework, Fianu et al. have posited a holistic approach of health determinants for investigating both individual and socio-environmental (contextual) risk factors of adult sedentary behavior living in Reunion Island with the aim to pave the way for future public heath interventions in the primary prevention of T2D. The topic is of public health relevance given the global epidemic of obesity and T2D, whose incidences peak in vulnerable communities like those observed in the southwestern Indian ocean (SWIO) region. The team from which it comes seems to have mastered this type of intervention, which warrants interesting perspectives for the transformation process into future actions (Fianu et al, Sante Publique 2017; Fianu et al, PLoS One 2016; Debussche et al, Diabetes Metabolism 2012). The paper is thorough and well-written. It carries and spreads very interesting methodological options that it deserves publication to shed light on the intricate links between social inequalities in health (SIH), physical and sport activity (PSA) and sedentary behaviors in the setting of T2D prevention. This notably because literature still lacks well-designed population-based studies from high-risk regions, which face multiple challenges best known under the concept of epidemiological transition. However, before reaching this achievement, the authors should consider minor compulsory and minor discretionary revisions including clearer presentation of study findings, language simplifications, and discussion of strategies in analysis. With an appropriate revision, given the abovementioned, the paper could even be eligible to PLoS Global Public Health, a very promising new PLOS journal of Public Health/Global Health with a strong focus on social inequalities in health.

Specific comments

Minor compulsory revision (mandatory)

Title and abstract

1(a). Indicate the study’s design with a commonly used term in the title or the abstract. Add “population-based” to study as it is a strength of the study.

The abstract is too narrative and should include quantitative results.

1(b). Page 2, line 31. Replace “territory” by department, which is more exact administratively.

Page 2, line 38. The authors must define the categories of interest of their multinomial outcome.

Page 2, lines 41 to 42. “… associated with sedentary or physical inactive behavior.” The formulation makes think the outcome is composite and binary while it is four categorial.

Page 2, line 44. I am not sure a research perspective has to be placed in the abstract.

Page 2, line49 to page 3, line 50. Rephrase the last sentence of the abstract as “reducing” may apply for lifestyle risk factors but not T2D primary prevention programs.

Page 2, line 49. “may suggest.” Why “may” ? The authors should better trust themselves and write “suggest” or “may foster” or something stronger like that.

Introduction

2. Background/rationale

Explain the scientific background and rationale for the investigation being reported.

Page 3, lines 58 to 59. Rephrase the sentence to distinguish among the determinants of chronic diseases (such as T2D), risk factors from mediators. For instances, writing “that are intrinsically linked to both the development of sedentary lifestyle and excessive caloric intake through metabolic disturbances (insulin resistance).” Diabetes should not be placed at the end of the sentence as it is one of the chronic diseases to what the authors refer to.

Page 3, lines 61.”is… and is…” Two times is for the same subject, T2D. Rephrase the sentence starting with Type 2 diabetes (T2D),…[4, 5].”

Page 3, line 73 to page 4, line 74. Does the author use social inequalities for social inequalities of health? If yes, replace social inequalities by SIH to avoid the truism as it seems evident that differences in income and unfavorable living and working conditions define social inequalities.

3. Objectives

State specific objectives, including any prespecified hypotheses.

The authors should be more explicit on their hypotheses. We guess they are presented in the long sentence preceding the objectives page 4, lines 82 to 84. They could use a formulation like “Given the abovementioned blablabla, we hypothesize … .” They should also mention and define physical and sport activities as physical activity/inactivity is included in their multinomial four- category outcome. For readers not familiar to the literature about physical activity and health, it may be difficult to distinguish sedentary behavior and absence of physical activity.

Methods

4. Study design

Present key elements of study design early in the paper

5. Setting

Describe the setting, locations, and relevant dates, including periods of recruitment, exposure, follow-up, and data collection.

Page 5, line 98. Define the acronym ERPPS in the text and makes clear that the study reports this study as it is published formerly in a study report for public health stakeholders, academics and funders but not as a scientific publication.

Page 5, line 98. “was collected” is abrupt and let imagine it these are findings of a new study disconnected from the ERPPS study. Better use “had been” to let see, the data had been previously collected.

Page 5, line 101. Write “aged 15 or over” as it is really “≥” and not “>”.

Page 5, line 107. Define the acronym “Fideli” in French in italics. Acronym should be defined in French and English translations added when necessary.

Page 5, line 112. Define the categories of the level of income to see how it was categorized. It will make more intelligible the number of possible strata in the random sampling.

7. Variables

Clearly define all outcomes, exposures, predictors, potential confounders, and effect modifiers. Give diagnostic criteria, if applicable

Lifestyle behaviors outcomes. This important paragraph should be rephrased and start from

the definition of the multinomial four-category outcome, followed by clearer definitions of each of its categories.

The authors clearly have focused their hypothesis and objectives on sedentary behavior and not physical activity. This paragraph should explain why physical activity was integrated in the outcome definition and note taken as a potential effect-modifier (or even as a potential confounder).

Page 6, line 134. Physical inactivity is a composite variable which combined four reported outcome variables.

Page 6, line 137. Remove “reporting” as it has been specified previously that data were self-reported.

Confounding factors. Page 7, lines 147 to 148. The sentence is confusing. How a health-related behavior like a sedentary behavior could confound sedentary behavior?

Factors of exposure. Page 7, line 158. Replace “threshold” by “categories” as we need three thresholds or cut-off to define four categories.

Page 7, line 160. I don’t think language spoken in the home may be defined as a psychosocial characteristic. Please explain.

Page 7, lines 179 to 182. Define the referent category as zero category and the categorization 0 to 4.

8. Data sources/ measurement

For each variable of interest, give sources of data and details of methods of assessment (measurement). Describe comparability of assessment methods if there is more than one group.

Page 6, line 124. Which ecological data were compiled?

9. Bias

Describe any efforts to address potential sources of bias

10. Study size

Explain why this study did not require sample size pre-specification (instead of how the study size was arrived at).

11. Quantitative variables

Explain how quantitative variables were handled in the analyses. If applicable, describe which groupings were chosen and why?

12. Statistical methods

12 (a). Describe all statistical methods, including those used to control for confounding.

12 (b). Describe any methods used to examine subgroups and interactions.

It should be explained why physical activity was integrated into the multinomial four-category outcome as it precludes study physical activity as an effect-modifier or a confounder.

12 (c). Explain how missing data were addressed. This paragraph page 9, lines 205 to 215 is

The paragraph page 9, lines 205 to 212 is difficult to read for non-statistician readers and has little impact on the understanding the article. I would shorten it and merely explain that there were multiple imputations.

Page 9, line 200. “The working framework.” Working could be replaced by operating or something like that.

Ethical considerations. Page 10, lines 234 to 235. Explain why minors 15 to 17 were not asked to provide their oral consent along with that of their parents.

Results

13. Participants

13 (a).Page 11, line 241. Remove the “with” and reformulate using a formulation like “which represented” or something like that.

14. Descriptive data

14 (a). Give characteristics of study participants (eg demographic, clinical, social) and information on exposures and potential confounders.

Page 11, lines 247 to 251. Although, Table 1 is a descriptive table, were there some comparisons between the study sample and census data. When percentages differed it could have shed light on potential selection biases? (even though in this case differences are significant due to statistical power.

Page 16, lines 273 to 274. Most participants possessed an educational degree. This could have been a source of selection bias. Explain and specify if there was or there wasn’t a selection bias and how it was accounted if it had an impact on study findings.

Page 16, lines 274 to 275. Define a positive perception.

Page 16, line 281. What for students?.

14 (b). Indicate number of participants with missing data for each variable of interest

15. Outcome data

15 (c). Cross-sectional study—Report numbers of outcome events or summary measures

16. Main results

16 (a). Unadjusted estimates could be placed as supporting information(supplemental online tables). Make clear which confounders were adjusted for and why they were included.

It is inappropriate and very confusing to report a single analysis on three different tables. Tables 2 to 4 should be merged into a single table presenting the M3 final model distinguishing each outcome category (Y1 in place of M1, Y2 in place of M2, as study findings that are interpreted are only those of M3 models. Results of nested models M1 and M2 models which are not interpreted or very simply in short sentences, should be presented as supporting information and supplemental online tables.

Tables 2 to 4. With respect to the prespecified hypotheses, why not having taken “final secondary school diploma or above as referent category?

16 (b). Report category boundaries when continuous variables were categorized

16 (c). If relevant, consider translating estimates of relative risk into absolute risk for a meaningful time period

17 Other analyses

Report other analyses done—eg analyses of subgroups and interactions, and sensitivity analyses.

Page 16, lines 264 to 267. Explain why results have not been presented at least for some microregions that shaped or contrasted the overall study findings.

Page 23, line 323. Present crude unadjusted analyses like the primary analysis: M1 with Y1,Y2 and Y3 in columns, M2 with Y1, Y2, and Y3 in columns.

Page 23, line 319. “Fewer poor neighborhoods” means literally in French “Moins de pauvres voisinages.” I think the authors mean “Less poor” as specified in the Methods.

Discussion

18. Key results

Summarise key results with reference to study objectives.

Why explaining only significant results? Please explain. Some expected findings may have not been found due to beta risk (lack of power), but this would interest researchers to know why.

Page 23, lines 331 to 332. “associated to sedentary behavior, taking into account physical activity status.” This formulation would better apply with physical status taken as an effect-modifier (interaction term) or a confounder. Read above and if not done, explain theses options were not retained for the primary analysis.

Page 24, lines 336 to 337. “Many studies reported the consequences on physical activity levels and sedentary.” The authors posit that sedentary behavior and physical inactivity are different things. Why amalgaming here? Please, detail other study findings here to support your statement.

Page 24, line 349. “Consistent”. I am not sure this term is appropriate here. The socio-economic is the most consistent risk factor for what? Please, detail.

Page 24, lines 357 to 358. Specify here that a “better integration of communication on physical activity content” is needed if physical activity may decrease the consequences of sedentary behaviors, which is not consistent across studies, as the authors pinpoint elsewhere.

19. Limitations

Discuss both strengths and limitations of the study, taking into account sources of bias or imprecision, maybe strengths prior to limitations. Discuss both direction and magnitude of any potential bias.

20. Interpretation

Give a cautious overall interpretation of results considering objectives, limitations, multiplicity of analyses, results from similar studies, and other relevant evidence.

Page 25, line 342. There is a misinterpretation here. Write “less poor” neighborhoods as the protective effective effect discussed here applies for the fourth category of the variable which indicates a less disadvantaged neighborhood (the fifth category indicating the wealthier advantaged neighborhood).

Page 25, line 368. If possible, provide here a reference and place it after areas.

Page 25, lines 367 to 370. The sentence is not so clear. Explain what is an endogenous psychosociological mechanism.

Page 25, lines 381 to 383. The authors state that the “inhibition of some protective health-related attitudes are found in the most deprived neighborhoods”. I guess this is an interpretation but where is the study finding on which it is based on?

Page 26, line 387; I am not sure the term “intervention” is suitable here. The authors refer to multiple public health actions. Qualify the intervention as multifaceted or better use program, for instance.

21. Generalisability

Discuss the generalisability (external validity) of the study results.

Page 27, line 421. “institutionalized accommodation.” Does the authors mean long-term care facility?

Page 27, line 423. What represent the people placed in institutions on Reunion island? Are they different or representative of the general population? If not and if there are numerous, could they introduce a selection bias?

Page 27, line 424. “The accelerometer gold standard”. Is a study in nuclear physics? Reformulate with a coma and a more appropriate designation of the gold standard method.

Page 28, line 442. How a health risk could be refined? Give more precisions because it is not crystal clear.

Other information

22. Funding

Give the source of funding and the role of the funders for the present study and, if applicable, for the original study on which the present article is based.

References

Translate the French references that are accessible in Pubmed into English and write them into brackets[] as done in PubMed.

Minor discretionary corrections (request an author’s reply without obligation of edits)

Title and abstract

1(b). Page 2, line 30. I am not sure that the prevalence is the target of primary prevention programs. Maybe incidence would be more appropriate. Prevalence decreases as a reduction of incidence.

Page 2, line 34. “to highlight… in order to …. to alleviate…” Too much “to”in the same sentence. Replace some of these by grammatically correct formulations. “aiming at + -ing”.

Page 2, line 47. “Strengthening individuals.” Does the authors mean enpowerment?

Page 3, lines 50 to 51. Material exposures is vague. It could be detailed.

Introduction

2. Background/rationale

Page 3, line 66. The PIMA could be named in the text as it is well known reminiscent example of the concept of disadvantaged communities regarding T2D.

Page 3, line 71. Add the acronym (SWIO) for southwestern Indian ocean and “region” after and specify it belongs to the WHO African region. This is important as Reunion island share social characteristics of both well developed and low-or-middle income countries.

Page 3, line 76. Too many wars are waged in the world. Please use a peaceful vocabulary. I do not think a combat could be waged against a probability as there is no ethics in preemptive (or preventive) wars. Rephrase using more intelligible words.

Page 3, line 81. I am not sure the results of reference #17 could be used for such generalization and the findings of a local study on Reunion island could be translated to all territories with aging populations. Complete the sentence stating this as an example with a formulation like “as a demographic study from Reunion island suggests”.

Page 3, line 87. “Strengthening individuals.” Does the authors mean enpowerment?

3. Objectives

State specific objectives, including any prespecified hypotheses.

Page 3, lines 91 to 92. “to highlight… in order to …. to alleviate…” Too much “to”in the same sentence. Replace some of these by grammatically correct formulations. “aiming at + -ing”.

Methods

7. Variables

Confounding factors. Page 7, line 145. Age categories have been categorized differently than for sampling. It could be explained why.

8. Data sources/ measurement

Page 5, line 119. “during the study period” would be better placed upstream in the sentence, after “were gathered;”

12. Statistical methods

12 (a). Page 8, line 190. Remove the “the” before four categories.

Ethical considerations. I see no concern to use “concerning” twice, however it not elegant.

Results

14. Descriptive data.

Page 16, line 273. People aged 15 to 44 years could be characterized as young adults here and further, the age ranged placed in parentheses.

Discussion

19. Limitations

20. Interpretation

Page 26, line 396. Fianu et al. Replace by “we” or by ”our group.“

21. Generalisability

Discuss the generalisability (external validity) of the study results.

Page 26, line 407. “The research perspective was.” A research perspective is sensed to occur in the future. Please replace “was” by the present, the future or the conditional (is, will be, or would be…).

Page 27, line 409. “At the frontiers of…” Replace by “at the borderline” or “at the edge”, better appropriate here.

Other information

References should be placed at the end of the sentences.

Page 9, line 199. Ref#23 should be placed at the end of the sentence.

Page 9, line 211. Ref#26 should be placed at the end of the sentence.

Page 24, line 349. Ref#25 should be placed at the end of the sentence.

Page 27, line 413. Ref#22 should be placed at the end of the sentence.

Page 28, line 446. Ref#23 should be placed at the end of the sentence.

Reviewer #2: Review of the manuscript entitled „Determinants of adult sedentary behavior for the primary prevention of diabetes in historically disadvantaged communities: a representative cross-sectional study from Reunion Island” (PONE-D-23-43256)

I read the manuscript offered for review with pleasure and interest. The study presents the results of original research, to the best of my knowledge the reported results have not been published elsewhere. The study is described by the authors with adequate statistical analysis and at a good technical level, in sufficient detail. The conclusions are well presented, the article is written in a comprehensible manner. English is generally sound, only minor revision. The research complied with all relevant standards for experimental ethics and research integrity. The article adheres to appropriate reporting guidelines and community standards for data availability.

I have the following questions and comments regarding the manuscript:

Introduction:

- The text discusses the impact of socio-economic changes on health behavior and type 2 diabetes (T2D). Although the connections are clear, the logical process of the text should be made clearer so that readers can follow the argument more easily. Showing the connections, for example how socio-economic changes lead to a higher prevalence of T2D, may require a more detailed explanation.

- The information about the island of Réunion is detailed and relevant, but the history of the text does not make it clear why this specific location is important. It would be worthwhile to emphasize more why this particular community is being studied and how it differs from other communities in a similar socio-economic situation.

- A more precise definition of terms such as "contextual influences" and "macro- or meso-economic and social factors" could help readers better understand the content of the text. The text mentions the island's historical context (slavery and indentured labor), but could also explain their effects on current health inequalities in more detail.

Material and methods:

- When using the phrase "Identical to the nationwide study of PSA (ENPPS-2020)", it would be useful to briefly summarize the purpose and methods of the ENPPS-2020 study so that readers can better understand the relationship between the two studies.

- The details of the sampling methods are thorough, but the term "one-level systematic random sampling stratified by the micro-region" may require an explanation as to exactly how the sampling took place.

- Mentioning the unequal sampling probability is important, but it would be helpful to provide further explanation of how this is handled in the analysis (e.g., by weighting).

- Mention of “positive perception of personal and parental history of PSA” as a potential confounder is important, but it is unclear how this variable was measured and what effect it may have had on the results.

Results:

- Regarding the sentence "In total, 2,010 adults from Reunion Island participated in the study with a participation rate of 49%", the question is how can the participation rate affect the reliability of the results in the study?

- During the presentation of the results, it would be good to provide a further explanation of the significance and interpretation of the confidence interval at the sentence "Over half the participants had a sedentary behavior (53.9%, 95%CI: [51.1 to 56.7])".

- In the case of the results for administrative micro-regions ("inhabitants of the northern region had the highest prevalence estimate for both sedentary/inactive (15.0%) and sedentary/active (46.1%)"), the differences are significant, but what factors can influence them?

- Regarding the generalizability of the results, it would be worthwhile to make further comments on how other regions can utilize the findings of the study.

Discussion:

- The transition from the effects of COVID-19 to the analysis of the socio-economic background is sudden. It would be worthwhile to provide a smoother transition, for example by connecting the two topics with how socio-economic background influenced individuals' responses during COVID-19.

- In addition to the cited statistical data (e.g. "one adult out of two" and "one adult out of five"), it would be worthwhile to provide exact percentages so that the reader can interpret the ratios more easily.

- The term "sedentary and active profile" needs clarification, since "sedentary" and "active" are opposite concepts. Perhaps the term "sedentary and physically active periods" would be more appropriate.

- The relationship between higher health literacy and physical activity mentioned by Rey et al (2023) deserves a more detailed explanation. For example, how higher health literacy can improve the effectiveness of health communication and how it can help promote physical activity.

- It is sometimes difficult to follow cause-and-effect relationships between the statements in the discussion. For example, the causes of physical activity inequalities and the mechanisms leading to them are not fully understood. The term "endogenous psychosocial mechanism on the stress perception" would require more explanation so that the reader can understand how this can lead to a sedentary lifestyle.

- The distinction between "exogenous" and "endogenous" factors is useful, but the examples in the text are not always clear. For example, "abandoning physical activity due to the COVID-19 pandemic" is presented as a behavioral factor, but it is not clear how it fits into the exogenous category.

- Among the intervention proposals is the strategy of "proportionate universalism", but it is not clear how this can be implemented in practice. More detailed explanations and concrete examples would be needed.

- The sentence "The understanding of individual and socio-environmental determinants..." is too general and not specific enough. It would be worth clarifying what specific interventions are possible with such knowledge.

- The "neighborhood deprivation level indicator" is specific to the island of Réunion, but it is not clear why and how it differs from indicators used elsewhere. A more detailed explanation would help the reader understand the differences.

- After the comment "multivariate analyzes did not include all the environmental factors", it would be worthwhile to clarify what specific environmental factors are missing and how they may affect the results.

6. PLOS authors have the option to publish the peer review history of their article (what does this mean?). If published, this will include your full peer review and any attached files.

Reviewer #1: **Yes: **PATRICK GERARDIN

Reviewer #2: No

---

## [Author Response · Author response to Decision Letter 0]

19 Jul 2024

Response to Editor Comments

We are grateful for the review, which helped us to improve the quality of the manuscript. We carefully addressed all points and modified the manuscript accordingly.

Sincerely,

Adrian Fianu, corresponding author

JOURNAL REQUIREMENTS:

Response 1: 

In order to apply the PLOS ONE's style requirements, we’re still using the recommended templates:

- Title, author, affiliations formatting guidelines (April 2021 version)

- Manuscript body formatting guidelines (April 2021 version)

We replaced the filename of “S1 File” with “S1 Appendix” and “S2 File” with “S2 Appendix”.

2. Please state what role the funders took in the study. 

Response 2: 

Since the funders had no role, we amended our statement as recommended:

"This research was funded by the European Regional Development Fund (PO IV 2014-2020), the Regional Council of La Reunion and the French State. The funders had no role in study design, data collection and analysis, decision to publish, or preparation of the manuscript."

We included this amended Role of Funder statement in our cover letter, to let the PLOS ONE’s staff change the online submission form on our behalf.

3. We note that Figure 3 in your submission contain [map/satellite] images which may be copyrighted. All PLOS content is published under the Creative Commons Attribution License (CC BY 4.0), which means that the manuscript, images, and Supporting Information files will be freely available online, and any third party is permitted to access, download, copy, distribute, and use these materials in any way, even commercially, with proper attribution. For these reasons, we cannot publish previously copyrighted maps or satellite images created using proprietary data, such as Google software (Google Maps, Street View, and Earth).

Response 3: 

We supplied a replacement background map from OpenStreetMap ® that complies with the CC BY 4.0 license. Please see full explanations on OpenStreetMap here:

https://www.openstreetmap.org/copyright/en

We also checked copyright information on the replacement figure (© openstreetmap contributors) and updated the figure caption with source information. Please see both the new Figure 3 and its caption.

In addition, provided that the Editor agree, we suggest removing the ERPPS study questionnaire (S1 Appendix), which seems to be a confidential IPSOS © document (see the questionnaire pages’ bottom).

Other information:

- On the manuscript's first page, we had to update the author's third affiliation by separating it into two parts: 3 Espace-Dev, IRD, Univ La Réunion, Saint Denis, La Réunion, France. 4 Espace-Dev, Univ La Réunion, Saint Denis, La Réunion, France.

- Given the requests of Reviewer # 1 for the abstract, we had to delete two entire sentences to not exceed 300 words (Please see the new abstract).

- On the S2 Appendix, we had to update one data source URL (because the former was no longer available).

- For convenience, we added the STROBE Checklist for Cross-sectional studies at the end of the responses given for Reviewer # 1 comments.

- Be careful in the responses given to the reviewers, page and line numbers refer to the Revised Manuscript with Track Changes, except for the STROBE Checklist referring to the revised and clean manuscript for convenience.

* * *

Response to Reviewers Comments

REVIEW COMMENTS TO THE AUTHOR:

Reviewer # 1: 

We appreciated the positive assessment of our study in the general comment and noticed that Reviewer # 1 used the STROBE checklist for observational studies (https://www.equator-network.org/reporting-guidelines/strobe/) for structuring his report. Within this framework, our responses considered specific Reviewer’s comment(s) under/after each STROBE item (we put them in Italics to better identify them). We assumed STROBE items without the Reviewer’s comment were not applicable or already presented in the text and did not expect a response from the authors. In addition, we completed the STROBE Checklist for Cross-sectional studies at the end of the responses given for Reviewer # 1.

Be careful in the following responses page and line numbers refer to the Revised Manuscript with Track Changes, except for the STROBE Checklist referring to the revised and clean manuscript for convenience.

Specific comments / Minor compulsory revision (mandatory):

Point 1: 

Title and abstract 1(a). Indicate the study’s design with a commonly used term in the title or the abstract. 

Add “population-based” to study as it is a strength of the study. 

Response 1: 

Fixed in manuscript (page 1, lines 2-3). 

Point 2: 

The abstract is too narrative and should include quantitative results.

Response 2: 

First, we included a prevalence 95% confidence interval for both sedentary and physical inactivity behaviors (page 3, lines 42-44).

Second, to better describe significant associations between lifestyle behaviors outcomes and factors of exposure, we added significant P-values from the complete full-adjusted regression models (M3) (page 3, lines 45-47). However, to not exceed 300 words in the abstract, it was not possible to put the corresponding odds ratios and their confidence intervals estimated from the same M3 models. In addition, we had to delete two entire sentences to not exceed the 300-word threshold (Please see the new abstract pages 3-4).

Point 3: 

1(b). Page 2, line 31. Replace “territory” by department, which is more exact administratively. 

Response 3: 

Fixed in manuscript (page 3, line 33).

Point 4: 

Page 2, line 38. The authors must define the categories of interest of their multinomial outcome.

Response 4: 

Done in manuscript (page 3, lines 41-42).

Point 5: 

Page 2, lines 41 to 42. “… associated with sedentary or physical inactive behavior.” The formulation makes think the outcome is composite and binary while it is four categorial.

Response 5: 

We suggest to re-use the categories introduced in Point 4. Please see (page 3, line 48).

Point 6: 

Page 2, line 44. I am not sure a research perspective has to be placed in the abstract.

Response 6: 

To not exceed 300 words in the abstract, we had to delete the sentence including the terms “research perspective” (Please see the new abstract pages 3-4). Thus, the amendment has become obsolete.

Point 7: 

Page 2, line49 to page 3, line 50. Rephrase the last sentence of the abstract as “reducing” may apply for lifestyle risk factors but not T2D primary prevention programs. 

Response 7: 

“enhancing” (page 4, line 57) seems a better public health perspective to qualify T2D primary prevention programs, as stated in the Conclusion of our manuscript (page 37).

Point 8: 

Page 2, line 49. “may suggest.” Why “may” ? The authors should better trust themselves and write “suggest” or “may foster” or something stronger like that.

Response 8: 

Thank you for this relevant advice, which empowers our propositions. As a consequence, we deleted the word “may” (page 4, line 56).

Point 9: 

2. Background/rationale.Explain the scientific background and rationale for the investigation being reported.

Page 3, lines 58 to 59. Rephrase the sentence to distinguish among the determinants of chronic diseases (such as T2D), risk factors from mediators. For instances, writing “that are intrinsically linked to both the development of sedentary lifestyle and excessive caloric intake through metabolic disturbances (insulin resistance).” Diabetes should not be placed at the end of the sentence as it is one of the chronic diseases to what the authors refer to.

Response 9: 

Fixed in manuscript (page 4, lines 65-67).

Point 10: 

Page 3, lines 61.”is… and is…” Two times is for the same subject, T2D. Rephrase the sentence starting with Type 2 diabetes (T2D),…[4, 5].”

Response 10: 

Fixed in manuscript (page 4, lines 69-70).

Point 11: 

Page 3, line 73 to page 4, line 74. Does the author use social inequalities for social inequalities of health? If yes, replace social inequalities by SIH to avoid the truism as it seems evident that differences in income and unfavorable living and working conditions define social inequalities.

Response 11: 

We did not use “social inequalities” for social inequalities in health (SIH). In this statement, our purpose was to describe the social context of Reunion Island independent of the health state of the population.

We respectfully disagree with the comment second part (second sentence in Point 11). Indeed, SIH are also underpinned by many other social determinants than living and working conditions, which are significant causes of SIH but not the only ones. For instance, according to a widely accepted classification, the upper-stream and may be most influential determinants of health are those related to general socio-economic, cultural and environmental conditions, including economic strategies, tax policies, trade, and environmental agreements between countries (Dahlgren and Whitehead, 2021).

Point 12: 

3. Objectives

State specific objectives, including any prespecified hypotheses.

The authors should be more explicit on their hypotheses. We guess they are presented in the long sentence preceding the objectives page 4, lines 82 to 84. They could use a formulation like “Given the abovementioned blablabla, we hypothesize … .” 

Response 12: 

As recommended, we introduced more explicitly the study’s hypothesis (page 6, lines 104-105).

Point 13: 

They should also mention and define physical and sport activities as physical activity/inactivity is included in their multinomial four- category outcome. For readers not familiar to the literature about physical activity and health, it may be difficult to distinguish sedentary behavior and absence of physical activity.

Response 13: 

“Physical and sport activities (PSA)” was first mentioned at the end of the second paragraph of the Introduction section. Due to the need not to unbalance the Introduction section (with a long and detailed text), we decided to develop this concept through both the WHO’s academic definition and the available proxy chosen for our analysis in the Materials and Methods section. Please see the Lifestyle behaviors outcomes section (pages 9-10, lines 190-198).

To introduce the difference and complementarity between sedentary behavior and physical inactivity, we added a statement just after the sedentary behavior definition (page 6, lines 101-102).

Point 14:

Methods

4. Study design

Present key elements of study design early in the paper

5. Setting

Describe the setting, locations, and relevant dates, including periods of recruitment, exposure,

follow-up, and data collection.

Page 5, line 98. Define the acronym ERPPS in the text and makes clear that the study reports this

study as it is published formerly in a study report for public health stakeholders, academics and

funders but not as a scientific publication.

Response 14: 

These study reporting precisions were added in page 7, lines 123-124, 127-128.

Point 15:

Page 5, line 98. “was collected” is abrupt and let imagine it these are findings of a new study

disconnected from the ERPPS study. Better use “had been” to let see, the data had been previously

collected.

Response 15: 

Fixed in manuscript (page 7, line 124).

Point 16:

Page 5, line 101. Write “aged 15 or over” as it is really “≥” and not “>”.

Response 16: 

Fixed in manuscript (page 6, lines 121-122).

Point 17:

Page 5, line 107. Define the acronym “Fideli” in French in italics. Acronym should be defined in

French and English translations added when necessary.

Response 17: 

We added the “Fideli” French definition (page 7, lines 144-145). The text already gave the English translation: “a demographic listing (for tax purposes) of accommodation and individuals”.

Point 18:

Page 5, line 112. Define the categories of the level of income to see how it was categorized. It will

make more intelligible the number of possible strata in the random sampling.

Response 18: 

Done in manuscript (page 8, line 150).

Point 19:

7. Variables

Clearly define all outcomes, exposures, predictors, potential confounders, and effect modifiers.

Give diagnostic criteria, if applicable

Lifestyle behaviors outcomes. This important paragraph should be rephrased and start from

the definition of the multinomial four-category outcome, followed by clearer definitions of each of

its categories.

Response 19: 

Done in manuscript (pages 8-10, lines 167-198).

Point 20:

The authors clearly have focused their hypothesis and objectives on sedentary behavior and not

physical activity. This paragraph should explain why physical activity was integrated in the outcome

definition and note taken as a potential effect-modifier (or even as a potential confounder).

Response 20: 

Thank you for this significant methodological comment. It helped us to better justify our lifestyle behaviors outcomes. Actually, given the study design (a cross-sectional survey), it is not possible to consider that physical activity status (measured at the time of the survey) could precede the occurrence of sedentary behavior (measured at the same time) as it should be the case for a confounder or an effect-modifier. Therefore, the physical activity status is part of the outcome (see Figure 1) and is not a confounder or an effect-modifier.

This explanation is consistent with the French National Observatory on Physical Activity and Sedentariness (ONAPS) recommendation. Please see online:

- Figure 1 at: 

https://onaps.fr/les-definitions/

- And also:

https://lnkd.in/dNv3jpbJ

In our manuscript, to avoid confusion, we updated the title (page 1), short title (page 1), abstract (pages 3-4, lines 31, 35-36) and main text, including “physical inactivity” in the study rationale (page 6, lines 101-102, 107) and objective (line 114).

Point 21:

Page 6, line 134. Physical inactivity is a composite variable which combined four reported outcome

variables.

Response 21: 

Fixed in manuscript (page 9, line 194).

Point 22:

Page 6, line 137. Remove “reporting” as it has been specified previously that data were selfreported.

Response 22: 

Fixed in manuscript (page 10, line 196).

Point 23:

Confounding factors. Page 7, lines 147 to 148. The sentence is confusing. How a health-related

behavior like a sedentary behavior could confound sedentary behavior?

Response 23: 

Thank you for this comment. It was a mistake. We updated the statement (page 11, lines 203-204).

Point 24:

Factors of exposure. Page 7, line 158. Replace “threshold” by “categories” as we need three

thresholds or cut-off to define four categories.

Response 24: 

We replaced the term “threshold” with “classes” (page 11, line 214) as BMI was considered an ordinal variable.

Point 25:

Page 7, line 160. I don’t think language spoken in the home may be defined as a psychosocial

characteristic. Please explain.

Response 25: 

In Reunion Island, Creole is the native spoken language, whereas French is the official language taught at school and used for administrative purposes. The language spoken in the home at the age of five (in three categories: Only Creole/Only French/French and Creole/Other languages) was considered by our team as a proxy of both literacy (including health literacy) and community belonging. According to this assumption, we classified this factor within the broad category of psychosocial determinants of health.

To clarify, we stated our assumption (page 11, lines 217-218).

Point 26:

Page 7, lines 179 to 182. Define the referent category as zero category and the categorization 0 to

4.

Response 26: 

We respectfully disagree with this request. In accordance with the INSEE typology defining the large neighborhood deprivation groups (1, 2, 3, 4, and 5), we chose to apply the published figures (Besson 2018). In this way, the reader will be able to match the INSSE publication and ours without difficulty. This convention has no impact on the referent category for the multinomial logistic regression model.

Point 27:

8. Data sources/ measurement

For each variable of interest, give sources of data a

---

## [Decision Letter · Decision Letter 1]

29 Jul 2024

Determinants of adult sedentary behavior and physical inactivity for the primary prevention of diabetes in historically disadvantaged communities: a representative cross-sectional population-based study from Reunion Island

PONE-D-23-43256R1

Dear Dr. Fianu,

We’re pleased to inform you that your manuscript has been judged scientifically suitable for publication and will be formally accepted for publication once it meets all outstanding technical requirements.

Kind regards,

Attila Csaba Nagy

Academic Editor

PLOS ONE

Additional Editor Comments (optional):

Reviewers' comments:

Reviewer's Responses to Questions

**Comments to the Author**

1. If the authors have adequately addressed your comments raised in a previous round of review and you feel that this manuscript is now acceptable for publication, you may indicate that here to bypass the “Comments to the Author” section, enter your conflict of interest statement in the “Confidential to Editor” section, and submit your "Accept" recommendation.

Reviewer #1: All comments have been addressed

Reviewer #2: All comments have been addressed

2. Is the manuscript technically sound, and do the data support the conclusions?

Reviewer #1: Yes

Reviewer #2: Yes

3. Has the statistical analysis been performed appropriately and rigorously? 

Reviewer #1: Yes

Reviewer #2: Yes

4. Have the authors made all data underlying the findings in their manuscript fully available?

Reviewer #1: Yes

Reviewer #2: Yes

5. Is the manuscript presented in an intelligible fashion and written in standard English?

Reviewer #1: Yes

Reviewer #2: Yes

6. Review Comments to the Author

Reviewer #1: Manuscript Number: PONE-D-23-43256-R1.

Title: Determinants of adult sedentary behavior for the primary pre of diabetes in historically disadvantaged communities : a representative cross-sectional population-based study from Reunion island.

Article Type: Research article

General comment

The authors have addressed all my remarks the best they can. I have no additional claim.

Specific comments

Abstract

Page 2, line 39 and 40 Add percentages to the figures in 95% confidence intervals.

Introduction

Page 3, line 56. Write “have resulted in…2

Discussion

The authors should reintroduce the quotation of their IJERP paper as it provide clues on SIH and behaviour during the COVID-19 pandemic on the island of La Reunion

Reviewer #2: (No Response)

7. PLOS authors have the option to publish the peer review history of their article (what does this mean?). If published, this will include your full peer review and any attached files.

Reviewer #1: **Yes: **Patrick Gérardin

Reviewer #2: No

---

## [Editor Report · Acceptance letter]

5 Aug 2024

PONE-D-23-43256R1 

PLOS ONE

Dear Dr. Fianu, 

I'm pleased to inform you that your manuscript has been deemed suitable for publication in PLOS ONE. Congratulations! Your manuscript is now being handed over to our production team.

Kind regards, 

on behalf of

Dr. Attila Csaba Nagy 

Academic Editor

PLOS ONE